# EFFICIENT ROBUST TRAINING VIA BACKWARD SMOOTHING

## ABSTRACT

Adversarial training is so far the most effective strategy in defending against adversarial examples. However, it suffers from high computational cost due to the iterative adversarial attacks in each training step. Recent studies show that it is possible to achieve Fast Adversarial Training by performing a single-step attack with random initialization. Yet, it remains a mystery why random initialization helps. Besides, such an approach still lags behind state-of-the-art adversarial training algorithms on both stability and model robustness. In this work, we develop a new understanding towards Fast Adversarial Training, by viewing random initialization as performing randomized smoothing for better optimization of the inner maximization problem. From this perspective, we show that the smoothing effect by random initialization is not sufficient under the adversarial perturbation constraint. A new initialization strategy, *backward smoothing*, is proposed to address this issue and significantly improves both stability and model robustness over single-step robust training methods. Experiments on multiple benchmarks demonstrate that our method achieves similar model robustness as the original TRADES method, while using much less training time ($\sim$3x improvement with the same training schedule).

## 1 INTRODUCTION

Deep neural networks are well known to be vulnerable to adversarial examples (Szegedy et al., 2013), *i.e.*, a small perturbation on the original input can lead to misclassification or erroneous prediction. Many defense methods have been developed to mitigate the disturbance of adversarial examples (Guo et al., 2018; Xie et al., 2018; Song et al., 2018; Ma et al., 2018; Samangouei et al., 2018; Dhillon et al., 2018; Madry et al., 2018; Zhang et al., 2019), among which robust training methods, such as adversarial training (Madry et al., 2018) and TRADES (Zhang et al., 2019), are currently the most effective strategies. Specifically, adversarial training method (Madry et al., 2018) trains a model on adversarial examples by solving a min-max optimization problem:

$$\min_{\boldsymbol{\theta}} \frac{1}{n} \sum_{i=1}^{n} \max_{\mathbf{x}_i' \in \mathcal{B}_\epsilon(\mathbf{x}_i)} L(f_{\boldsymbol{\theta}}(\mathbf{x}_i'), y_i), \tag{1.1}$$

where $\{(\mathbf{x}_i, y_i)\}_{i=1}^n$ is the training dataset, $f(\cdot)$ denotes the logits output of the neural network, $\mathcal{B}_\epsilon(\mathbf{x}_i) := \{\mathbf{x} : \|\mathbf{x} - \mathbf{x}_i\|_\infty \leq \epsilon\}$ denotes the $\epsilon$-perturbation ball, and $L$ is the cross-entropy loss.

On the other hand, instead of directly training on adversarial examples, TRADES (Zhang et al., 2019) further improves model robustness with a trade-off between natural accuracy and robust accuracy, by solving the empirical risk minimization problem with a robust regularization term:

$$\min_{\boldsymbol{\theta}} \frac{1}{n} \sum_{i=1}^{n} \left[ L(f_{\boldsymbol{\theta}}(\mathbf{x}_i), y_i) + \beta \max_{\mathbf{x}_i' \in \mathcal{B}_\epsilon(\mathbf{x}_i)} \mathrm{KL}\big(s(f_{\boldsymbol{\theta}}(\mathbf{x}_i)), s(f_{\boldsymbol{\theta}}(\mathbf{x}_i'))\big) \right], \tag{1.2}$$

where $s(\cdot)$ denotes the softmax function, and $\beta > 0$ is a regularization parameter. The goal of this robust regularization term (*i.e.*, KL divergence term) is to ensure the outputs are stable within the local neighborhood. Both adversarial training and TRADES achieve good model robustness, as shown on recent model robustness leaderboards[1] (Croce & Hein, 2020b; Chen & Gu, 2020).

---

[1] https://github.com/fra31/auto-attack and https://github.com/uclaml/RayS.

However, a major drawback lies in that both are highly time-consuming for training, limiting their usefulness in practice. This is largely due to the fact that both methods perform iterative adversarial attacks (*i.e.*, Projected Gradient Descent) to solve the inner maximization problem in each outer minimization step.

Recently, Wong et al. (2020) shows that it is possible to use single-step adversarial attacks to solve the inner maximization problem, which previously was believed impossible. The key ingredient in their approach is adding a random initialization step before the single-step adversarial attack. This simple change leads to a reasonably robust model that outperforms other fast robust training techniques, *e.g.*, Shafahi et al. (2019). However, it remains a mystery why random initialization is empirically effective. Furthermore, compared to state-of-the-art robust training models (Madry et al., 2018; Zhang et al., 2019), this approach still lags behind on model robustness.

In this work, we aim to understand the role of random initialization, as well as closing the robustness gap between adversarial training and Fast Adversarial Training (Fast AT) (Wong et al., 2020). We propose a new principle towards understanding Fast AT - that random initialization can be viewed as performing randomized smoothing for better optimization of the inner maximization problem. We demonstrate that the smoothing effect by random initialization is not sufficient under the adversarial perturbation constraint. By proposing a new initialization strategy, *backward smoothing*, which strengthens the smoothing effect within the $\epsilon$-perturbation ball, we present a new fast robust training method based on TRADES (Zhang et al., 2019). The resulting method significantly improves both stability and model robustness over the single-step version of TRADES (Zhang et al., 2019), while consuming much less training time ($\sim$ 3x improvement with the same training schedule).

## 2 RELATED WORK

There exists a large body of work on adversarial attacks and defenses. In this section, we only review the most relevant work to ours.

**Adversarial Attack** The concept of adversarial examples was first proposed in Szegedy et al. (2013). Since then, many methods have been proposed, such as Fast Gradient Sign Method (FGSM) (Goodfellow et al., 2015), and Projected Gradient Descent (PGD) (Kurakin et al., 2016; Madry et al., 2018). Later on, various attacks (Papernot et al., 2016; Moosavi-Dezfooli et al., 2016; Carlini & Wagner, 2017; Athalye et al., 2018; Chen et al., 2020; Croce & Hein, 2020a) were also proposed for better effectiveness or efficiency. There are also many attacks focused on different attack settings. Chen et al. (2017) proposed a black-box attack where the gradient is not available, by estimating the gradient via finite-differences. Various methods (Ilyas et al., 2018; Al-Dujaili & O'Reilly, 2020; Moon et al., 2019; Andriushchenko et al., 2019; Tashiro et al., 2020) have been developed to improve the query efficiency of Chen et al. (2017). Other methods (Brendel et al., 2018; Cheng et al., 2019; 2020) focused on the more challenging hard-label attack setting, where only the prediction labels are available. On the other hand, there is recent work (Croce & Hein, 2020b; Chen & Gu, 2020) that aims to accurately evaluate the model robustness via ensemble of attacks or effective hard-label attack.

**Robust Training** Many heuristic defenses (Guo et al., 2018; Xie et al., 2018; Song et al., 2018; Ma et al., 2018; Samangouei et al., 2018; Dhillon et al., 2018) were proposed when the concept of adversarial examples was first introduced. However, they are later shown by Athalye et al. (2018) as not truly robust. Adversarial training (Madry et al., 2018) is the first effective method towards defending against adversarial examples. In Wang et al. (2019), a new convergence quality criterion was proposed. Zhang et al. (2019) showed the trade-off between natural accuracy and robust accuracy. Wang et al. (2020) proposed to improve model robustness by better utilizing misclassified examples. Another line of research utilizes extra information (*e.g.*, pre-trained models (Hendrycks et al., 2019) or extra unlabeled data (Carmon et al., 2019; Alayrac et al., 2019)) to further improve robustness. Other work focuses on improving training efficiency, such as free adversarial training from Shafahi et al. (2019) and Fast AT from Wong et al. (2020) using single-step attack (FGSM) with random initialization. Li et al. (2020) proposed a hybrid approach for improving Fast AT which is orthogonal to ours. Andriushchenko & Flammarion (2020) proposed a new regularizer promoting gradient alignment. Yet, it is not focused on closing the robustness gap with state-of-the-arts.

**Randomized Smoothing** Duchi et al. (2012) proposed the randomized smoothing technique and proved variance-based convergence rates for non-smooth optimization. Later on, this technique was

applied to certified adversarial defenses (Cohen et al., 2019; Salman et al., 2019) for building robust models with certified robustness guarantees. In this paper, we are not targeting certified defenses. Instead, we use the randomized smoothing concept in optimization to explain Fast AT.

## 3 WHY RANDOM INITIALIZATION HELPS?

We aim to explain why random initialization in Fast AT is effective, and propose a new understanding that random initialization can be viewed as performing randomized smoothing on the inner maximization problem in adversarial training (Madry et al., 2018). Below, we first introduce the randomized smoothing technique (Duchi et al., 2012) in optimization.

It is well known from optimization theory (Boyd et al., 2004) that non-smooth objectives are generally harder to optimize compared with smooth objectives. In general, a smoother loss function allows us to use a larger step size while guaranteeing the convergence of gradient-based algorithms. Randomized smoothing technique (Duchi et al., 2012) was proposed based on the observation that random perturbation of the optimization variable can be used to transform the loss into a smoother one. Instead of using only the gradient at the original iterate, randomized smoothing proposes to randomly generate perturbed iterates and use their gradients for optimization procedure. More details are provided in Appendix A. Let us recall the inner maximization problem in adversarial training:

$$\max_{\boldsymbol{\delta} \in \mathcal{B}_\epsilon(\mathbf{0})} L(f_{\boldsymbol{\theta}}(\mathbf{x} + \boldsymbol{\delta}), y). \tag{3.1}$$

Here, $f_{\boldsymbol{\theta}}$ denotes a neural network parameterized by $\boldsymbol{\theta}$. In general, neural networks are non-smooth due to ReLU activations and pooling layers. This suggests that (3.1) can be difficult to solve, and using gradient descent with large step size can lead to divergence in the maximization problem. It also explains why directly using single-step projected gradient ascent without random initialization fails (Wong et al., 2020). Now, let us apply randomized smoothing to (3.1):

$$\max_{\boldsymbol{\delta} \in \mathcal{B}_\epsilon(\mathbf{0})} \mathbb{E}_{\boldsymbol{\xi} \sim U(-1,1)} L(f_{\boldsymbol{\theta}}(\mathbf{x} + \boldsymbol{\delta} + \epsilon \boldsymbol{\xi}), y), \tag{3.2}$$

where $\boldsymbol{\xi}$ is the perturbation vector for randomized smoothing, and $\boldsymbol{\delta}$ is the perturbation vector for later gradient update step (initialized as zero). Suppose we solve (3.2) in a stochastic fashion (*i.e.*, sample a random perturbation $\boldsymbol{\xi}$ instead of computing the expectation over $\boldsymbol{\xi}$), and using only one step gradient update. We can see that this reduces to the Fast AT formulation. This suggests that Fast AT can be viewed as performing stochastic single-step attacks on a randomized smoothed objective function which allows using larger step size. This explains why random initialization helps Fast AT as it makes the loss objective smoother and thus easier to optimize.

It is worth noting that Andriushchenko & Flammarion (2020) also provided an explanation of why Fast Adversarial Training works: random initialization reduces the magnitude of the perturbation and thus the network becomes more linear and fits better toward single-step attack. While we argue that the random initialization works as randomized smoothing for smoothing the inner maximization problem and makes it easier to solve. In fact, our argument is more general and can cover theirs, because if the loss function is approximately linear, then it will be very smooth, i.e., the second-order term in the Taylor expansion is very small.

## 4 PROPOSED APPROACH

### 4.1 DRAWBACKS OF THE RANDOM INITIALIZATION STRATEGY

Although Fast AT achieves much faster robust training compared with standard adversarial training (Madry et al., 2018), it exposes several major weaknesses. For demonstration, we exclude the additional acceleration techniques introduced in Wong et al. (2020) for accelerating the training speed (*e.g.,* mix-precision training, cyclic learning rate), and instead apply standard *piecewise learning rate decay* used in Madry et al. (2018); Zhang et al. (2019) with the decay point set at the 50-th and 75-th epochs.

**Performance Stability** As observed in Li et al. (2020), Fast AT can be highly unstable (*i.e.*, large variance in robust performance) when using traditional piecewise learning rate decay schedule. We argue that this is because Wong et al. (2020) utilized a drastically large attack step size ($10/255$, even larger than the perturbation limit $\epsilon$), which causes unstable training behavior.

Table 1: Model robustness comparison among Fast Adversarial Training, Adversarial Training, and TRADES, using ResNet-18 model on CIFAR-10 dataset.

| Method | Nat (%) | Rob (%) |
|---|---|---|
| Fast AT (avg. over 10 runs) | 84.58 | 44.52 |
| Fast AT (best over 10 runs) | 84.79 | 46.30 |
| AT (early-stop) | 82.36 | 51.14 |
| TRADES | 82.33 | 52.74 |

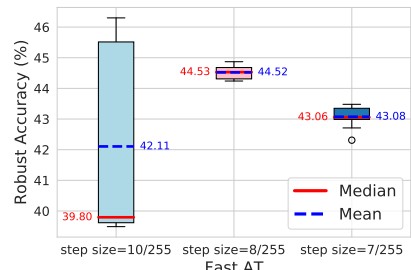

Figure 1: Comparison of Fast Adversarial Training performance with different step sizes.

To validate this, we run Fast AT on CIFAR-10 using ResNet-18 model (He et al., 2016) for 10 times with different step sizes. Note that we adopt early-stopping and record the best-performing model among 100 epochs. As shown in Figure 1, although the single-best robustness performance is obtained by using step size $10/255$, the variance is very high. Moreover, most trials lead to weak robust performance with a low average and median robust accuracy. On the other hand, we observe that when using step size $8/255$, model robustness is more stable and higher on average. Note that using a too small step size would by nature hurt model robustness. These observations suggest that Fast AT cannot achieve the best performance on robust performance and stability simultaneously.

**Potential for Robustness Improvement** Fast AT uses standard adversarial training (Madry et al., 2018) as the baseline, and can obtain similar robustness performance. However, later work (Rice et al., 2020) shows that adversarial training can cause the overfitting problem, while early stopping can largely improve robustness. Zhang et al. (2019) further achieves even better model robustness that is much higher than what Fast AT obtains. From Table 1, we observe that there exists an $8\%$ robust accuracy gap between Fast AT (average over 10 runs) and the best-performing TRADES model. Even for the best out of 10 trials, there is still a $6\%$ gap. This indicates that Fast AT is still far from optimal, and there is still big room for further robustness improvement.

## 4.2 A Naive Try: Randomized Smoothing for TRADES

As shown in Table 1, TRADES enjoys better model robustness compared with standard adversarial training. A naive attempt is to apply randomized smoothing to TRADES and see if this leads to better robustness than Fast AT. Let us recall the inner maximization formulation for TRADES:

$$\max_{\boldsymbol{\delta} \in \mathcal{B}_\epsilon(\mathbf{0})} \text{KL}\big(s(f_{\boldsymbol{\theta}}(\mathbf{x})), s(f_{\boldsymbol{\theta}}(\mathbf{x} + \boldsymbol{\delta}))\big). \tag{4.1}$$

Similarly, we can smooth this objective and solve the following objective instead:

$$\max_{\boldsymbol{\delta} \in \mathcal{B}_\epsilon(\mathbf{0})} \mathbb{E}_{\boldsymbol{\xi} \sim U(-1,1)} \text{KL}\big(s(f_{\boldsymbol{\theta}}(\mathbf{x})), s(f_{\boldsymbol{\theta}}(\mathbf{x} + \boldsymbol{\delta} + \epsilon\boldsymbol{\xi}))\big). \tag{4.2}$$

This leads to the same adversarial example formulation as using random initialization and then performing single-step projected gradient ascent. We refer to this strategy as Fast TRADES.

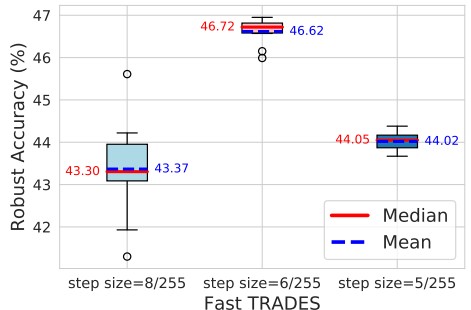

Figure 2: Comparison of Fast TRADES performance under different step sizes.

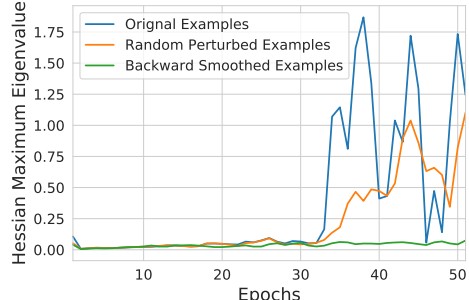

Figure 3: Hessian maximum eigenvalue comparison against training epochs.

We experiment with Fast TRADES using different step sizes, and find that its performance is sensitive to step size, similar to Fast AT. As shown in Figure 2, using a large step size of $8/255$ leads

to very low average and median robust accuracy. We notice that reducing the step size to $6/255$ yields a better average robust accuracy, which is also slightly higher than Fast AT. Nevertheless, the improvements here over Fast AT are not significant. This inspires us to study how to design a better strategy for more significant improvements.

Recall the results from Section 4.1. Applying overly-large step size in Fast AT and Fast TRADES can lead to unstable training with deteriorated robustness. This suggests that the randomized smoothing effect might not be strong enough (*i.e.*, the objective function is not smooth enough) to enable the use of a larger step size. However, unlike the general randomized smoothing setting, one of the special constraints in the adversarial setting is that random perturbation on the input vector is subject to the $\epsilon$-ball constraint, therefore cannot be too large. This means that we cannot further increase the smoothing effect by simply using larger random perturbations.

To further validate the claim that the random smoothing effect is not sufficient, we carefully study the loss smoothness under different smoothing techniques. Figure 3 shows the maximum eigenvalue of Hessian of the loss function at the original examples, randomly perturbed examples, and backward smoothed examples along the training trajectory until Fast TRADES obtains its best robustness (the 51st epoch). We observe that during the model training process, the randomly perturbed examples have overall smaller Hessian maximum eigenvalue[2] than that of original examples. This suggests that random smoothing indeed makes the loss function smoother. Moreover, the Hessian maximum eigenvalue under backward smoothing is much smaller than that under random smoothing, showing the insufficiency of the random smoothing techniques and the advantages of our proposed backward smoothing method.

### 4.3    BACKWARD SMOOTHING

Now we introduce our proposed method to address the above issue. The goal is to further boost the smoothing effect of randomized smoothing without violating the $\epsilon$-perturbation constraint. Note that if we are allowed to use larger random perturbations, we expect that $\text{KL}(s(f_{\boldsymbol{\theta}}(\mathbf{x})), s(f_{\boldsymbol{\theta}}(\mathbf{x}+u\boldsymbol{\xi})))$ will also be larger, meaning that the neural network output of the random initialization $f_{\boldsymbol{\theta}}(\mathbf{x}+u\boldsymbol{\xi})$ should be more different from the original output $f_{\boldsymbol{\theta}}(\mathbf{x})$ (as shown in Figure 4). This inspires us to generate the initialization point in a backward fashion. Specifically, let us denote the input domain $\mathbf{x} \in \mathbb{R}^d$ as the input space, and their corresponding neural network output $f_{\boldsymbol{\theta}}(\mathbf{x}) \in \mathbb{R}^c$ as the output space, where $c$ is the number of classes for the classifier. We first generate random points in the output space just as randomized smoothing does in the input space, *i.e.*, $f_{\boldsymbol{\theta}}(\mathbf{x}) + \gamma\psi$, where $\psi \sim U(-1, 1)$ is the random variable and $\gamma$ is a small number. Then we find the corresponding

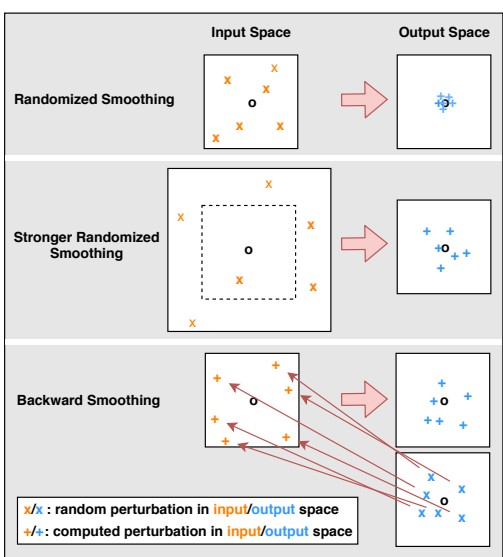

Figure 4: A sketch of our proposed method.

input perturbation in a backward fashion and use it as our initialization. An illustrative sketch of our proposed method is provided in Figure 4.

Now we formalize our proposed method in mathematical language. The key step in our proposed method is to find the input perturbation $\boldsymbol{\xi}$ such that:

$$f_{\boldsymbol{\theta}}(\mathbf{x}+\boldsymbol{\xi}) = f_{\boldsymbol{\theta}}(\mathbf{x}) + \gamma\psi. \tag{4.3}$$

In order to find the best $\boldsymbol{\xi}^*$ to satisfy (4.3), we turn to solve the following problem:

$$\boldsymbol{\xi}^* = \underset{\boldsymbol{\xi} \in \mathcal{B}_{\epsilon}(\mathbf{0})}{\arg\min} \text{KL}\big(s(f_{\boldsymbol{\theta}}(\mathbf{x}) + \gamma\psi), s(f_{\boldsymbol{\theta}}(\mathbf{x}+\boldsymbol{\xi}))\big). \tag{4.4}$$

---

[2]The smaller Hessian maximum eigenvalue, the smoother the loss function is.

Note that $\xi$ is initialized as a zero vector. For the sake of computational efficiency, we solve (4.4) using single-step PGD in practice. Then, similar to Wong et al. (2020), we use single-step gradient update for the inner maximization problem:

$$\boldsymbol{\delta}^* = \operatorname*{argmax}_{\boldsymbol{\delta} \in \mathcal{B}_\epsilon(\mathbf{0})} \mathrm{KL}\big(s(f_{\boldsymbol{\theta}}(\mathbf{x})), s(f_{\boldsymbol{\theta}}(\mathbf{x} + \boldsymbol{\delta} + \boldsymbol{\xi}^*))\big). \tag{4.5}$$

Finally, we update the neural network parameter $\boldsymbol{\theta}$ using stochastic gradients at the adversarial point $\mathbf{x} + \boldsymbol{\xi}^* + \boldsymbol{\delta}^*$. A summary of our proposed algorithm is provided in Algorithm 1. Note that the proposed Backward Smoothing seems also compatible with Adversarial Training. However, Adversarial Training does not contain terms using KL divergence loss, which may hinder its performance. We will show this empirically in Section 5.

---

**Algorithm 1** Backward Smoothing

1: **input:** The number of training iterations $T$, number of adversarial perturbation steps $K$, maximum perturbation strength $\epsilon$, training step size $\eta$, adversarial perturbation step size $\alpha$, regularization parameter $\beta > 0$;
2: Random initialize model parameter $\boldsymbol{\theta}_0$
3: **for** $t = 1, \ldots, T$ **do**
4:     Sample mini-batch $\{\mathbf{x}_i, y_i\}_{i=1}^m$ from training set
5:     Obtain $\boldsymbol{\xi}^*$ by solving (4.4)
6:     Obtain $\boldsymbol{\delta}^*$ by solving (4.5)
7:     $\boldsymbol{\theta}_t = \boldsymbol{\theta}_{t-1} - \eta/m \cdot \sum_{i=1}^m \nabla_{\boldsymbol{\theta}} \big[ L(f_{\boldsymbol{\theta}}(\mathbf{x}_i), y_i) + \beta \cdot \mathrm{KL}\big(s(f_{\boldsymbol{\theta}}(\mathbf{x}_i)), s(f_{\boldsymbol{\theta}}(\mathbf{x}_i + \boldsymbol{\xi}^* + \boldsymbol{\delta}^*))\big)\big]$
8: **end for**

---

We notice that Tashiro et al. (2020) proposed an attack which also samples diversified points in the output space. Yet their method is not focused on randomized smoothing and has a totally different formulation, and they are generating diversified output points for better attack efficiency rather than achieving better defense.

## 5 EXPERIMENTS

In this section, we empirically evaluate the performance of our proposed method. We first compare our proposed method with other robust training baselines on CIFAR-10, CIFAR100 (Krizhevsky et al., 2009) and Tiny ImageNet (Deng et al., 2009)[3] datasets. We also provide multiple ablation studies as well as robustness evaluation with state-of-the-art adversarial attack methods to validate that our proposed method provides effective robustness improvement.

### 5.1 EXPERIMENTAL SETTING

Following previous work on robust training (Madry et al., 2018; Zhang et al., 2019; Wong et al., 2020), we set $\epsilon = 0.031$ for all three datasets. In terms of model architecture, we adopt standard ResNet-18 model (He et al., 2016) for both CIFAR-10 and CIFAR-100 datasets, and ResNet-50 model for Tiny ImageNet. We follow the standard piecewise learning rate decay schedule as used in Madry et al. (2018); Zhang et al. (2019) and set decaying point at 50-th and 75-th epochs. The starting learning rate for all methods are set to 0.1, the same as previous work (Madry et al., 2018; Zhang et al., 2019). For all methods, we tune the models for their best robustness performances. For Adversarial Training and TRADES methods, we adopt 10-step iterative PGD attack with step size $2/255$ for both. For our proposed method, we set the backward smoothing parameter $\gamma = 1$. For robust accuracy evaluation, we typically adopt 100-step PGD attack with step size $2/255$. To ensure the validity of the model robustness improvement is not because of the obfuscated gradient (Athalye et al., 2018), we further test our method with current state-of-the-art attacks (Croce & Hein, 2020b; Chen & Gu, 2020).

### 5.2 PERFORMANCE COMPARISON WITH ROBUST TRAINING BASELINES

We compare the adversarial robustness of Backward Smoothing against standard Adversarial Training (Madry et al., 2018), TRADES (Zhang et al., 2019), as well as fast training methods such as

---

[3]We do not test on ImageNet dataset mainly due to that TRADES does not perform well on ImageNet as mentioned in Qin et al. (2019).

Fast AT (Wong et al., 2020) and our naive baseline Fast TRADES. We also compare with recently proposed Fast AT+ method (Li et al., 2020) that achieves high robustness with reduced training time.[4] Since our proposed backward smoothing initialization utilizes an extra step of gradient back-propagation, we also compare with Fast TRADES using 2-step attack for fair comparison.

Table 2: Performance comparison on CIFAR-10 using ResNet-18 model.

| Method | Nat (%) | Rob (%) | Time (m) |
|---|---|---|---|
| AT | 82.36 | 51.14 | 430 |
| Fast AT | **84.79** | 46.30 | **82** |
| Fast AT (2-step) | 83.21 | 49.91 | 127 |
| TRADES | 82.33 | **52.74** | 482 |
| Fast TRADES | 83.39 | 46.98 | 126 |
| Fast TRADES (2-step) | 83.51 | 48.78 | 164 |
| *Backward Smoothing* | 82.38 | 52.50 | 164 |

Table 3: Performance comparison on CIFAR-100 using ResNet-18 model.

| Method | Nat (%) | Rob (%) | Time (m) |
|---|---|---|---|
| AT | 55.22 | 28.53 | 428 |
| Fast AT | **60.35** | 24.64 | **83** |
| Fast AT (2-step) | 56.00 | 27.84 | 128 |
| TRADES | 56.99 | 29.41 | 480 |
| Fast TRADES | 60.26 | 21.33 | 126 |
| Fast TRADES (2-step) | 58.81 | 25.47 | 165 |
| *Backward Smoothing* | 56.96 | **30.50** | 164 |

Table 2 shows the performance comparison on the CIFAR-10 dataset using ResNet-18 model. Our Backward Smoothing method significantly closes the robustness gap between state-of-the-art robust training methods, achieving high robust accuracy that is almost as good as TRADES, while consuming much less ($\sim$ 3x) training time. Compared with Fast AT, Backward Smoothing typically costs twice the training time, yet achieving significantly higher model robustness. Our method also achieves similar performance gain against Fast TRADES. Note that even compared with Fast TRADES using 2-step attack and Fast AT using 2-step attack, which costs about the same training time as ours, our method still achieves a large improvement.

Table 3 shows the performance comparison on CIFAR-100 using ResNet-18 model. We can observe patterns similar to CIFAR-10 experiments. Backward Smoothing achieves slightly higher robustness compared with TRADES, while costing much less training time. Compared with Fast TRADES using 2-step attack and Fast AT using 2-step attack, our method also achieves a large robustness improvement with roughly the same training cost. Table 4 shows that on Tiny ImageNet using ResNet-50 model, Backward Smoothing also achieves sig-

Table 4: Performance comparison on Tiny ImageNet dataset using ResNet-50 model.

| Method | Nat (%) | Rob (%) | Time (m) |
|---|---|---|---|
| AT | 44.50 | 21.34 | 2666 |
| Fast AT | **49.58** | 18.56 | **575** |
| Fast AT (2-step) | 45.74 | 20.94 | 817 |
| TRADES | 47.02 | 21.04 | 2928 |
| Fast TRADES | 50.36 | 17.22 | 805 |
| Fast TRADES (2-step) | 46.92 | 19.26 | 1045 |
| *Backward Smoothing* | 46.68 | **22.32** | 1035 |

nificant robustness improvement over other single-step robust training methods.

## 5.3 Evaluation with State-of-the-art Attacks

To ensure that Backward Smoothing does not cause obfuscated gradient problem (Athalye et al., 2018) or presents a false sense of security, we further evaluate our method using state-of-the-art attacks, by considering two evaluation methods: ($i$) AutoAttack (Croce & Hein, 2020b), which is an ensemble of four diverse (white-box and black-box) attacks (APGD-CE, APGD-DLR, FAB (Croce & Hein, 2020a) and Square Attack (Andriushchenko et al., 2019)) to reliably evaluate robustness; ($ii$) RayS attack (Chen & Gu, 2020), which only requires the prediction labels of the target model (completely gradient-free) and is able to detect falsely robust models. It also measures another robustness metric, average decision boundary distance (ADBD), defined as examples' average distance to their closest decision boundary. ADBD reflects the overall model robustness beyond $\epsilon$ constraint. Both evaluations provide online robustness leaderboards for public comparison with other models.

We train our method with WideResNet-34-10 model (Zagoruyko & Komodakis, 2016) and evaluate via AutoAttack and RayS. Table 5 shows that under state-of-the-art attacks, Backward Smoothing still holds high robustness comparable to TRADES. Specifically, in terms of robust accuracy, Backward Smoothing is only 2% behind TRADES, while significantly higher than AT (Madry et al., 2018) and Fast AT (Wong et al., 2020). In terms of ADBD metric, Backward Smoothing achieves

---

[4]Since Li et al. (2020) does not have code released yet, we only compare with theirs in the same setting (combined with acceleration techniques) using reported numbers.

the same level of overall model robustness as TRADES, much higher than the other two methods. Note that the gap between Backward Smoothing and TRADES is larger than that in Table 2. We want to emphasize that this is not mainly due to the stronger attacks[5] but the fact that we are using larger model architectures. Intuitively speaking, larger models have larger capacities and may need stronger attacks to reach some dark spot in the area.

Table 5: Performance comparison with state-of-the-art robust models on CIFAR-10 evaluated by AutoAttack and RayS.

| Method | AutoAttack | RayS | |
| Metric | Rob (%) | Rob (%) | ADBD |
|---|---|---|---|
| AT | 44.04 | 50.70 | 0.0344 |
| AT (early-stop) | 49.10 | 54.00 | 0.0377 |
| Fast AT | 43.21 | 50.10 | 0.0334 |
| TRADES | **53.08** | **57.30** | **0.0403** |
| Fast TRADES | 43.84 | 52.05 | 0.0348 |
| Fast TRADES (2-step) | 48.20 | 54.43 | 0.0383 |
| *Backward Smoothing* | 51.13 | 55.08 | **0.0403** |

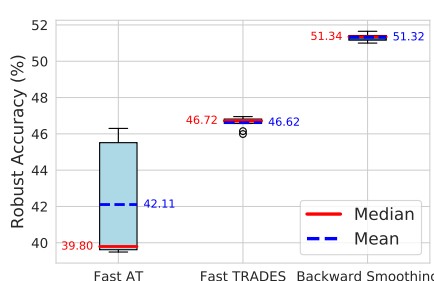

Figure 5: Robustness stability of different fast robust training methods.

## 5.4 STABILITY AND COMPATIBILITY

Figure 5 shows that Backward Smoothing is much more stable than Fast AT with much smaller variance. Compared with Fast TRADES, Backward Smoothing has achieved similar variance while obtaining much higher average model robustness. This demonstrates the superiority on robustness stability for Backward Smoothing method. We also wonder whether Backward Smoothing is compatible with Adversarial Training, *i.e.*, can we use a similar initialization strategy for improving Fast AT? We test this on CIFAR-10 using ResNet-18 model, and the resulting model achieves $45.53 \pm 0.35\%$ robust accuracy, improving the stability of Fast AT as well as the average robustness. However, the best run out of 10 trials does not achieve better robustness. We conjecture that the main reason for the deteriorated performance is the different choices of inner maximization loss for Adversarial Training (Cross-Entropy) and TRADES (KL divergence). Considering the random perturbation generated on the output space, the Cross-Entropy loss mainly focuses on the $y$-logit while KL divergence is closely related to all logits. This partially explains the above observations.

## 5.5 ABLATION STUDIES

We also perform a set of ablation studies to provide a more in-depth analysis on Backward Smoothing. Due to the space limit, here we present the sensitivity analysis on smoothing parameter $\gamma$ and the step size, and leave more ablation studies in the supplemental materials.

**Effect of** $\gamma$: We analyze the effect of $\gamma$ in Backward Smoothing by fixing $\beta$ and the attack step size. Table 6 summarizes the results. In general, $\gamma$ does not have a significant effect on the final model robustness; however, using too large or too small $\gamma$ would lead to slightly worse robustness. Empirically, $\gamma = 1$ achieves the best performance on both datasets.

Table 6: Sensitivity analysis of $\gamma$ on the CIFAR-10 and CIFAR-100 datasets using ResNet-18 model.

| Dataset | CIFAR-10 | | CIFAR-100 | |
| $\gamma$ | Nat (%) | Rob (%) | Nat (%) | Rob (%) |
|---|---|---|---|---|
| 0.1 | 82.43 | 52.13 | 56.62 | 29.34 |
| 0.5 | 82.53 | 52.34 | 56.95 | 29.85 |
| 1.0 | 82.38 | **52.50** | 56.96 | **30.50** |
| 2.0 | 82.29 | 52.42 | 56.16 | 29.88 |
| 5.0 | 81.50 | 52.32 | 56.10 | 429.83 |

Table 7: Sensitivity analysis of the attack step size on the CIFAR-10 and CIFAR-100 datasets using ResNet-18 model.

| Dataset | CIFAR-10 | | CIFAR-100 | |
| Step Size | Nat (%) | Rob (%) | Nat (%) | Rob (%) |
|---|---|---|---|---|
| 6/255 | 81.38 | 52.38 | 56.83 | 29.78 |
| 7/255 | 81.96 | 52.40 | 56.61 | 29.82 |
| 8/255 | 82.38 | **52.50** | 56.96 | **30.50** |
| 9/255 | 82.47 | 52.16 | 56.45 | 29.35 |
| 10/255 | 81.71 | 52.04 | 60.85 | 24.21 |
| 11/255 | 67.43 | 42.45 | 40.40 | 20.92 |
| 12/255 | 65.56 | 41.12 | 37.90 | 18.83 |

[5]We also tested the ResNet-18 models in Table 2 with AutoAttack and the gap between Backward Smoothing and TRADES is as small as $0.5\%$.

**Effect of Attack Step Size**: To verify the effect of attack step size, we fix $\gamma$ and $\beta$. From Table 7, we can observe that different from single-step robust training methods, Backward Smoothing achieves similar robustness with slightly smaller step size, while the best performance is obtained with step size $8/255$. This suggests that we do not need to pursue overly-large step size for better robustness as in Fast AT. This avoids the stability issue in Fast AT.

## 5.6 COMBINING WITH OTHER ACCELERATION TECHNIQUES

Aside from random initialization, Wong et al. (2020) also adopts two additional acceleration techniques to further improve training efficiency with a minor sacrifice on robustness performance: cyclic learning rate decay schedule (Smith, 2017) and mix-precision training (Micikevicius et al., 2017). We show that such strategies are also applicable to Backward Smoothing. Table 8 provides the results when these acceleration techniques are applied. We can observe that both work universally well for all methods, significantly reducing training time (in comparison with Table 2). Yet it does not alter the conclusions that Backward Smoothing achieves similar robustness to TRADES with much less training time. Also when compared with the recent proposed Fast AT+ method, Backward Smoothing achieves higher robustness and training efficiency. Note that the idea of Fast AT+ method is orthogonal to ours and we can also adopt such hybrid approach for further reduction on training time.

Table 8: Performance comparison on CIFAR-10 using ResNet-18 model combined with cyclic learning rate and mix-precision training.

| Method | Nat (%) | Rob (%) | Time (m) |
|---|---|---|---|
| AT | 81.48 | 50.32 | 62 |
| Fast AT | 83.26 | 45.30 | **12** |
| Fast AT+ | 83.54 | 48.43 | 28 |
| TRADES | 79.64 | **50.86** | 88 |
| Fast TRADES | **84.40** | 45.96 | 18 |
| Fast TRADES (2-step) | 81.37 | 47.56 | 24 |
| *Backward Smoothing* | 78.76 | 50.58 | 24 |

## 6 CONCLUSIONS

In this paper, we propose a new understanding towards Fast Adversarial Training by viewing random initialization as performing randomized smoothing for the inner maximization problem. We then show that the smoothing effect by random initialization is not enough under adversarial perturbation constraint. To address this issue, we propose a new initialization strategy, Backward Smoothing. The resulting method closes the robustness gap to state-of-the-art robust training methods and significantly improves model robustness over single-step robust training methods.

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

## A    RANDOMIZED SMOOTHING

Randomized smoothing technique (Duchi et al., 2012) was originally proposed for solving convex non-smooth optimization problems. It is based on the observations that random perturbation of the optimization variable can be used to transform the loss into a smoother one. Instead of using only $L(\mathbf{x})$ and $\nabla L(\mathbf{x})$ to solve

$$\min L(\mathbf{x}),$$

randomized smoothing turns to solve the following objective function, which utilizes more global information from neighboring areas:

$$\min \mathbb{E}_{\boldsymbol{\xi} \sim U(-1,1)} L(\mathbf{x} + u\boldsymbol{\xi}), \tag{A.1}$$

where $\boldsymbol{\xi}$ is a random variable, and $u$ is a small number. Duchi et al. (2012) showed that randomized smoothing makes the loss in (A.1) smoother than before. Hence, even if the original loss $L$ is non-smooth, it can still be solved by stochastic gradient descent with provable guarantees.

## B    ADDITIONAL ABLATION STUDIES

In this section, we conduct additional ablation studies to provide a comprehensive view to the Backward Smoothing method.

### B.1    THE EFFECT OF $\beta$

We conduct the ablation studies to figure out the effect of $\beta$ in Backward Smoothing method by fixing $\gamma$ and the attack step size. Table 9 shows the experimental results. Similar to what $\beta$ does in TRADES (Zhang et al., 2019), here in Backward Smoothing, $\beta$ still controls the trade-off between natural accuracy and robust accuracy. We observe that with a larger $\beta$, natural accuracy keeps decreasing and the best robustness is obtained with $\beta = 10.0$.

Table 9: Sensitivity analysis of $\beta$ on CIFAR-10 and CIFAR-100 datasets using ResNet-18 model.

| Dataset | CIFAR-10 | | CIFAR-100 | |
|---|---|---|---|---|
| $\beta$ | Nat (%) | Rob (%) | Nat (%) | Rob (%) |
| 2.0 | 84.87 | 46.46 | 62.22 | 24.83 |
| 4.0 | 84.58 | 50.01 | 59.03 | 27.58 |
| 6.0 | 83.96 | 51.65 | 57.46 | 28.66 |
| 8.0 | 82.48 | 51.88 | 57.51 | 29.38 |
| 10.0 | 82.38 | **52.50** | 56.96 | **30.50** |
| 12.0 | 81.63 | 52.38 | 56.46 | 29.95 |

### B.2    DOES BACKWARD SMOOTHING ALONE WORKS?

To further understand the role of Backward Smoothing in robust training, we conduct experiments on using Backward Smoothing alone, i.e., only use Backward Smoothing initialization but do not perform gradient-based attack at all. Table 10 and Table 11 show the experimental results. We can observe that Backward Smoothing as an initialization itself only provides a limited level of robustness (not as good as single-step attack). This is reasonable since the loss for Backward Smoothing does not directly promote adversarial attacks. Therefore it only serves as an initialization to help single-step attacks better solve the inner maximization problems.

Table 10: Performance of using Backward Smoothing alone on CIFAR-10 dataset using ResNet-18 model.

| Method | Nat (%) | Rob (%) |
|---|---|---|
| Fast AT | 84.79 | 46.30 |
| Fast TRADES | 84.80 | 46.25 |
| Backward Smoothing Alone | 69.87 | 39.26 |

Table 11: Performance of using Backward Smoothing alone on CIFAR-100 dataset using ResNet-18 model.

| Method | Nat (%) | Rob (%) |
|---|---|---|
| Fast AT | 60.35 | 24.64 |
| Fast TRADES | 60.22 | 19.40 |
| Backward Smoothing Alone | 43.47 | 18.51 |

### B.3    MORE EXPERIMENTS FOR BACKWARD SMOOTHING USING MULTIPLE RANDOM POINTS

We also conducted extra experiments using multiple random points for the Backward Smoothing method. As can be seen from Table 12, a single random point already leads to similar performance

Table 12: Sensitivity analysis on the number of random points used in Backward Smoothing on the CIFAR-10 dataset using ResNet-18 model.

| # RandPoints | Rob (%) | Time (m) |
|:---:|:---:|:---:|
| 1 | 52.50 | 164 |
| 2 | 52.67 | 204 |
| 5 | 52.70 | 316 |
| 10 | 52.73 | 510 |

Table 13: Performance comparison on CIFAR-10 dataset using ResNet-18 model ($\epsilon = 16/255$).

| Method | Nat (%) | Rob (%) | Time (m) |
|:---|:---:|:---:|:---:|
| AT | 62.76 | **32.03** | 425 |
| Fast AT | 53.72 | 20.12 | **89** |
| TRADES | 62.09 | 28.63 | 470 |
| Fast TRADES | 56.55 | 17.47 | 137 |
| Fast TRADES (2-step) | 53.36 | 19.11 | 167 |
| *Backward Smoothing* | 63.47 | 25.04 | 164 |

as multiple random points but saves more time. Note that our target is to improve the efficiency of adversarial training, therefore, we only use a single random point for randomized smoothing in our proposed method.

### B.4 MORE EXPERIMENTS FOR LARGER $\epsilon = 16/255$

We also conducted experiments testing the larger $\epsilon = 16/255$ case. As can be seen from Table 13, Backward Smoothing still achieves significant improvements over other single-step adversarial training algorithms despite that Adversarial training actually obtains better results compared to TRADES. This verifies the effectiveness of the proposed method.

