# OpenReview forum: "Efficient Robust Training via Backward Smoothing"
_ICLR.cc/2021/Conference — Reject_

### Official Review · AnonReviewer3 · 2020-10-15

**Rating:** 6
**Confidence:** 4

**Review:**

Summary: This paper seeks to reduce the training time of TRADES adversarial training. It tries to understand fast adversarial training methods (single-step adversary methods) by viewing the random initialization of the adversarial perturbation in the PGD steps as randomized smoothing, making the inner maximization (the adversary) easier. They find empirically that this smoothing isn't enough for vanilla fast adversarial training with the TRADES objective (in the sense that it doesn't improve much beyond fast AT after switching out the objective), requiring large step sizes that bring instability to training. Then, they aim to improve fast TRADES by their proposed algorithm, "backward smoothing", which first perturbs the output of the model and then solves the inverse problem to find the corresponding input that would make this perturbation. The intuition is to make sure that the smoothing averages over a variety of output values to get a stronger smoothing effect. This seems to give similar results to TRADES while cutting down the training time significantly (only two inner PGD steps).

Strengths:
- The paper is pretty clearly written, the results are impressive in terms of cutting down the training time of TRADES about 4x, and there are a substantial number of experiments, including fair comparisons with 2-step fast TRADES and on strong attacks.
- Sometimes, the robust accuracy is even better than TRADES, but the clean accuracy is almost always a bit worse.
- They use many steps of PGD (100 steps) for robust accuracy evaluation.
- The method is relatively simple (similar mechanics to PGD), and uses only 2 inner gradient steps. It seems to give more stable results than fast AT or fast TRADES, although most of the stability gain seems to come from switching to the TRADES objective (Fig 5).

Weaknesses:
- My main concern is the backward smoothing method seems general but only really works for TRADES (and not PGD AT). The intuitions that motivate the method don't seem special to TRADES, so it would be good to have some understanding of this phenomenon. Can the intuition given in the paper be verified somehow? Also, other papers have shown that tweaking PGD-AT (such as using early stopping) allows it to get just as good or better results than TRADES, so it's not clear that we should necessarily prefer a method based on TRADES.
- The authors mention the natural way to increase smoothing, which would be to use larger random perturbations. They argue that because of the norm constraint, we cannot increase the perturbations. However, even with the norm constraint it's clear that larger perturbations would not be the same as small perturbations, and could plausibly have a different effect. This point would be more convincing if there were some experiments just using larger perturbations for random initialization.
- Fast TRADES seems to be at a different point of the tradeoff curve between natural accuracy and robust accuracy, and perhaps it responds differently to the TRADES hyperparameter. From Table 2,3 etc. , we can see that Fast TRADES tends to improve natural accuracy and be worse than robust accuracy than TRADES. Could tuning the hyperparameter for Fast TRADES be enough to emulate the TRADES result?

Other/clarifications:
- Can the authors clarify how $\xi$ is initialized in eq 4.4?
- Is there any insight to what differences between backward smoothing and TRADES are exposed with a stronger attack?

---

> ### Public Comment · ~Abhay_Yadav1 · 2020-11-16
> **Should be compared with PGD-2 (as uses only 2 inner gradient steps) ?**
>
> Does it make sense to compare performance with PGD-2 (2 pgd steps) as both methods will have the same computation cost (assuming "backward smoothing" can be applied on top of FAST-AT) ?

---

> > ### Author Response · Authors · 2020-11-17
> > **We further compared with Fast-AT (2-step)**
> >
> > Please see our response to Reviewer 1, the answer to the third question.

---

> ### Author Response · Authors · 2020-11-17
> **Authors’ Response to Reviewer 3**
>
> Thank you for your insightful comments. Below, we provide detailed responses to your questions.
>
>
> Q1: My main concern is the backward smoothing method seems... not clear that we should necessarily prefer a method based on TRADES.
>
> A1: In Table 1, we have shown that even with early stopping, TRADES still outperforms PGD-AT under the same training conditions (though early stopping does make the difference much smaller). We also added PGD-AT with early-stopping in Table 5 and it also suggests similar results.  Moreover, in fact, both adversarial training and TRADES are not state-of-the-art robust training methods (see AutoAttack leaderboard), while the current best method is still based on TRADES. Our method can be easily implemented based on those new methods and achieve better robustness. Therefore, there is actually a strong reason to prefer to use TRADES as our basis. Also we want to clarify that our method does not only work for TRADES. It can be applied to PGD AT and achieve marginal improvements with an improved stability as discussed in Section 5.4.
>
> Q2: The authors mention the natural way to ...if there were some experiments just using larger perturbations for random initialization.
>
> A2: Thank you for your suggestion, but we may not be able to perform your described experiment. First, we have to use the same norm constraints, otherwise, the results are not directly comparable under different threat models. If so, using larger perturbation does not make it better as we need to project it back to satisfy the norm constraints and this may cause the random perturbations all lying on the surface of the epsilon ball. To further address your concern, we conduct experiments according to your suggestion anyway by enlarging the random perturbation range to [0, 2*eps] and then continue with Fast AT. This leads to a robustness of 43.23% on CIFAR-10 dataset, which is even slightly worse than the original Fast AT algorithm.
>
> Q3: Fast TRADES seems to be at a different point of the tradeoff ...tuning the hyperparameter for Fast TRADES be enough to emulate the TRADES result?
>
> A3: Thank you for your suggestion, we have comprehensively tuned the result for Fast TRADES (step sizes have also been tuned).
>
> | Beta | Nat   | Rob   |
> |------|-------|-------|
> | 2    | 87.34 | 41.55 |
> | 4    | 85.98 | 42.02 |
> | 6    | 84.80 | 45.91 |
> | 8    | 83.39 | 46.98 |
> | 10   | 83.03 | 46.46 |
>
> Indeed, Fast TRADES could correspond to slightly shifted trade-off points as can be seen from the above table. However, the best results here are still far from the robustness achieved by our Backward Smoothing algorithm (52.50%) and there is still large gap compared to the standard TRADES method.
>
> Q4: Can the authors clarify how ξ is initialized in eq 4.4?
>
> A4: We are sorry for the confusion. ξ is initialized as zero vector in eq 4.4.
>
> Q5: Is there any insight to what differences between backward smoothing and TRADES are exposed with a stronger attack?
>
> A5: Note that aside from different attacks, we actually adopt larger models (WideResNet) in Table 5’s results. And we believe that this is the major factor for the difference here rather than the different attacks. Intuitively speaking, larger architectures possess larger capacities and have more complicated decision boundaries. Therefore, may need stronger attacks to reach the dart spots in the area. On the other hand, we also test AutoAttack on smaller ResNet18 models used in Table 2, and the following results suggest that the difference by using different attacks are indeed marginal.
>
> | Methods            | PGD-100 | AutoAttack |
> |--------------------|---------|------------|
> | Backward Smoothing | 52.50   | 48.36      |
> | TRADES             | 52.74   | 48.86      |

---

> > ### Comment · AnonReviewer3 · 2020-11-22
> > **Response to authors**
> >
> > Thanks to the authors for running the additional experiments. Many of my minor concerns are alleviated, and I believe there is strong empirical evidence for this procedure to allow for training TRADES quickly (replace Fast TRADES).
> >
> > However the overall picture remains that the procedure as given is not necessarily something very general (that works across many robust training methods) that you'd expect from a general framework such as randomized smoothing. As other reviewers have mentioned, the motivation in terms of randomized smoothing is a bit weak in its connection to the actual method and the experiments do not clearly show that the given explanations about smoothing are indeed the main contributor to the improvement.  Further investigation into this aspect would make elucidate this aspect. I will keep my score as I think the empirical evidence is pretty strong for faster TRADES, but I think the motivations for the method and ablations for understanding should be presented and conducted more precisely than its current form.

---

> > > ### Author Response · Authors · 2020-11-23
> > > **Thank you for your response**
> > >
> > > We are glad to hear that most of your concerns have been addressed. We still want to argue that the motivation of our method is actually quite clear. We provided a new understanding of Fast Adversarial Training through the lens of randomized smoothing (a technique widely used for optimizing non-smooth functions in the optimization community). Indeed, Reviewers 1,2,4 mentioned an alternative explanation in (Andriushchenko & Flammarion, 2020) that reducing the magnitude of the perturbation helps with Fast Adversarial Training by making the loss more linear. Nevertheless, we would like to point out that our smoothing perspective is more essential and general than the linear-loss based interpretation in  (Andriushchenko & Flammarion, 2020), since linear loss is perhaps the simplest special case of smooth loss functions. In terms of the ablations for understanding smoothing’s role, we have just added a new experiment replacing old Figure 3 in the revision. Specifically, we show the maximum eigenvalue of Hessian of the loss function at the original examples, randomly perturbed examples, and backward smoothed examples along the training trajectory. The result confirms that randomized smoothing indeed helps make the loss function smoother yet our proposed backward smoothing technique makes the loss much smoother compared to randomized smoothing. This confirms the motivation of our proposed method.

---

### Official Review · AnonReviewer1 · 2020-10-27
**Good empirical results, theoretical explanation to improve**

**Rating:** 5
**Confidence:** 5

**Review:**

The paper proposes Backward Smoothing to close the gap in terms of robustness between standard multi-step and fast (one or two steps) adversarial training (AT). In particular, at high level, given a network $f$, a point $x$ and its logits $f(x)$ it suggests to first sample a random perturbation $\psi$ in the logits space around $f(x)$ and second computing, via one step of gradient descent, $\xi^*$ so to minimize the KL-divergence between the softmax values of $f(x) + \psi$ and $f(x + \xi^*)$. Then $x + \xi^*$ is used as starting point for one-step AT. The rationale behind the scheme is that the random step commonly added to $x$ before AT might be not sufficient to achieve a smoothing effect on the loss function and then an effective optimization.

Pros
1. The proposed method is simple and easily integrated into the TRADES training scheme.
2. The empirical results are strong, as Backward Smoothing  achieves better robustness than other fast AT versions and is more efficient that multi-step AT (including TRADES) with little loss in robustness. Also, the models are thoroughly tested with many white- and black-box attacks.

Cons
1. The theoretical explanation of why random initialization helps and the connection to randomized smoothing (Sec. 3) is not very convincing in my opinion. Randomized smoothing aims at estimating better directions for the gradient step, but it's not clear whether this is the case when using a single random point (this would need to be experimentally validated). An alternative explanation of the role of random initialization is presented in (Andriushchenko & Flammarion, 2020), which would be worth discussing.
2. The proposed method seems to share some similarities with ODI (Tashiro et al., 2020), which also proposes a scheme to find good starting points for PGD-based adversarial attacks optimizing some (randomly sampled) loss in the logits space. While in that work there's no focus on adversarial training, the overall idea looks similar, and I think the authors should comment on this.

Other comments and questions
1. In the comparison to other methods, I think for completeness it'd be good to show also Fast AT with 2 steps, since it has the same computational cost as Backward Smoothing and Fast AT performs better than Fast TRADES especially on CIFAR-100 and Tiny ImageNet. Additionally, including other techniques for fast adversarial training (Shafahi et al., 2019, Andriushchenko & Flammarion, 2020) might give a more complete picture.
2. In Table 5, the gap between Backward Smoothing and TRADES is larger than what reported in Table 2. Is this because of the larger architecture or just the evaluation with stronger attacks (or something else)?
3. How does the proposed technique behave for larger $\epsilon$? Is it still able to achieve results similar to multi-step AT (usually for larger $\epsilon$ the gap between single- and multi-step AT becomes even larger)?

Overall, as mentioned above, the paper presents strong empirical results which support the proposed method. However, the justification for the method as presented is not particularly convincing and should be better validated. Also, a discussion of the similarities with prior work seems needed.

Tashiro et al., "Diversity can be Transferred: Output Diversification for White- and Black-box Attacks"

---
Update after rebuttal

After reading the authors' response and the other reviews, I think the paper has quite clear pros and cons.

The experimental results (especially at $\epsilon=8/255$) are strong and the underlying idea of finding a good starting point for single step adversarial training makes sense to me (see reply below).

On the other side, the initial (and partially current) explanation relying on randomized smoothing given by the authors is not convincing, in particular when discussing the role of the random step in the success of Fast AT. The new experiments provided in the revision which rather analyze the smoothness of the starting point found by Backward Smoothing look like a much better explanation of the success of the proposed method (note that although the overlap of terminology I don't see a direct interpretation of the smoothness of the loss function at some specifically crafted point in terms of randomized smoothing). This should be more thoroughly analyzed and commented, which would consist in a quite major update of the paper in my opinion.

The current version doesn't provide an exhaustive motivation and analysis of the proposed method (in any direction, randomized smoothing or others), although the revision improves in this sense. Then, although I appreciate the good empirical results and I'm still in favor of the proposed method, I have to lower the score. I encourage the authors to clarify the weaknesses of the paper, which might result in a better understanding of what's needed for a successful fast adversarial training.

---

> ### Author Response · Authors · 2020-11-17
> **Authors’ Response to Reviewer 1**
>
> Thank you for your insightful comments. Below, we provide detailed responses to your questions.
>
> Q1: The theoretical explanation of why random initialization ...An alternative explanation of the role of random initialization is presented... which would be worth discussing.
>
> A1:  First, please refer to our response to all reviewers regarding the relationship with the pointed paper. Second, thank you for your suggestion, we have added extra experiments of using multiple random points. As can be seen from the following table, a single random point already leads to similar performance but saves more time. Note that our target is to improve the efficiency of adversarial training, therefore, we only use a single random point for randomized smoothing.
>
> | # random points | Robust Accuracy (PGD-100) | Time (min) |
> |-----------------|---------------------------|------------|
> | 1               | 52.50                     | 164        |
> | 2               | 52.67                     | 204        |
> | 5               | 52.70                     | 316        |
> | 10              | 52.73                     | 510        |
>
>
> Q2: The proposed method seems to share some similarities ... and I think the authors should comment on this.
>
> A2: Thank you for pointing it out. We will comment and discuss this paper in the revision.
>
> Q3: In the comparison to other methods, I think for completeness it'd be good to show also Fast AT with 2 steps, ...might give a more complete picture.
>
> A3: Thank you for your suggestion, we have added experiments for Fast AT with 2 steps. It achieves a robustness of 49.91% and 27.84% for CIFAR-10 and CIFAR-100 dataset on the ResNet-18 model, which still falls behind our Backward Smoothing method (52.50%/30.50%). We will update the tables in the revision. Compared with other baselines, Fast AT paper has already shown its advantage over (Shafahi et al., 2019) in their original paper. For (Andriushchenko & Flammarion, 2020), according to the policy, we are not required to compare with the recent work published less than 2 months from submission, but we will try to add a comparison later.
>
> Q4: In Table 5, the gap between Backward Smoothing... the larger architecture or just the evaluation with stronger attacks (or something else)?
>
> A4: We believe that the larger architecture is the main factor that influences the final robustness. Larger architectures intuitively have larger capacities and may need stronger attacks to reach some dark spot in the area. On the other hand, we also test AutoAttack on smaller ResNet18 models used in Table 2, and the following results suggest that the difference by using different attacks are indeed marginal.
>
> | Methods            | PGD-100 | AutoAttack |
> |--------------------|---------|------------|
> | Backward Smoothing | 52.50   | 48.36      |
> | TRADES             | 52.74   | 48.86      |
>
> Q5: How does the proposed technique behave...gap between single- and multi-step AT becomes even larger)?
>
> A5: Thank you for your suggestion, we have added experiments with eps=16/255 for CIFAR-10 data.
>
> | Method               | Nat (%) | Rob (%) | Time (min) |
> |----------------------|---------|---------|------------|
> | AT                   | 62.76   | 32.03   | 425        |
> | Fast AT              | 53.72   | 20.12   | 89         |
> | TRADES               | 62.09   | 28.63   | 470        |
> | Fast TRADES          | 56.55   | 17.47   | 137        |
> | Fast TRADES (2-step) | 57.22   | 19.28   | 167        |
> | Backward Smoothing   | 63.47   | 25.04   | 164        |
>
> We can see that Backward Smoothing still achieves significant improvements over other single-step adversarial training algorithms although the gap between multi-step adversarial training methods is indeed larger.

---

> > ### Comment · AnonReviewer1 · 2020-11-24
> > **Response to Authors**
> >
> > Thanks for the response (and sorry for the late reply).
> >
> > As mentioned in the initial review, I think the results and the evaluation are solid and further strengthened by the additional experiments in the revision.
> >
> > Also the method seems reasonable to me, and the new analysis about the magnitude of the eigenvalues of the Hessian is actually quite insightful to me and a more convincing justification of the scheme than what argued initially and in the first part of the paper (Section 3). In fact, in my understanding, a starting point with smoother loss gives a more informative gradient (to maximize the loss in Eq. 4.5) and then a single step is sufficient. Is this the meaning of such experiment for the authors?
> >
> > Still, I can't relate this to the principle behind randomized smoothing, which is averaging over different samples. In A.1 the authors use multiple points but it seems that all are chosen with backward smoothing (right?). In Section 3 the claim seems to be that to better solving the problem in Eq. 3.1 a better estimation of the gradient is needed, which can be achieved via averaging the gradients at different random points, but in practice even a single random initial point already suffices. This can be summarized as the gradient at a random point in the Lp-ball is more informative than that at the original point.
> > However, this is not validate and Fig. 3 in my opinion suggests that the difference (in smoothness) of the loss at the original and random point is not clear (especially given the large variance across epochs). Could the authors comment on this?
> >
> > Overall, I think the proposed methods is effective and better supported with the additional analysis in the revision.

---

> > > ### Author Response · Authors · 2020-11-25
> > > **Thank you for your reply**
> > >
> > > 1. “in my understanding, a starting point with smoother loss gives a more informative gradient (to maximize the loss in Eq. 4.5) and then a single step is sufficient. Is this the meaning of such experiment for the authors?”
> > > - Yes, you are right about our new experiment.
> > >
> > > 2. “Still, I can't relate this to the principle behind randomized smoothing...can the authors comment on this?”
> > > - Randomized smoothing is originally defined in the form of expectation over random perturbation. In practice, in order to approximate the exception, one can use sample average over multiple points. Nevertheless, in practice, even using a single random point to approximate the expectation often suffices to implement randomized smoothing. This is analogous to using stochastic gradient descent to solve a stochastic optimization problem defined in the form of expectation over random functions. As we all know, stochastic gradient descent with mini-batch size equal to 1 often works well enough to solve the stochastic optimization problem. The only caveat is that when the minibatch size is small, there could be chances that the stochastic gradient direction is very different from the true gradient direction. But in expectation (or with high probability), the stochastic gradient descent still converges. Actually, Figure 3 exactly demonstrates a similar phenomenon, that is, the loss for random perturbed points is in general smoother than the original points, while oscillations do happen sometimes due to the use of a single random point. We hope this clears your concern.

---

### Official Review · AnonReviewer2 · 2020-10-27
**Good experimental results but I don't agree with the motivation of the method (+ results on larger eps should be shown)**

**Rating:** 5
**Confidence:** 5

**Review:**

**Summary:**
The paper proposes a new algorithm for performing fast adversarial training. The proposed algorithm consists in solving the inner maximization problem in the following way: first, one takes a step of projected gradient descent (PGD) wrt an auxiliary loss (motivated by the idea of backward smoothing), and then one takes another step of PGD wrt the original loss function which is the KL divergence as suggested by the TRADES paper. The authors argue that taking two steps of PGD wrt different losses leads (perhaps, surprisingly) to higher robustness compared to taking two steps of PGD wrt the same original loss function.


**Pros:**
- Better robustness of models when computations are bounded with only 2 steps of PGD to solve the inner maximization problem.
- Thorough robustness evaluation which includes the recent AutoAttack.
- Ablation studies that help to better understand the influence of the two main hyperparameters of the proposed method.
- The paper is clearly written although I would rather disagree with the justifications for the proposed method (see below).


**Cons:**
- The explanations from Section “Why random initialization helps?” are not convincing because randomness is just one part of the solution proposed by Wong et al. (2020) to solve the inner maximization procedure. Another part is reducing the step size from 2\*eps to 1.25\*eps which effectively reduces the norm of the perturbation compared to standard FGSM as discussed in a recent related paper Andriushchenko & Flammarion (2020). Therefore, taking into account the crucial effect of the step size as shown in Fig. 3 in Wong et al. (2020), I don’t think there is a reason to believe that randomness *alone* has any positive influence on the solution of the inner maximization problem. This is my main concern as the whole paper is built on the connection to randomized smoothing. I would be open to raise the score if you can show that randomization alone (with a single $\xi$ or even multiple $\xi$ to approximate the expectation in Eq. (3.2)) with the full step size (2\*eps) alone can prevent catastrophic overfitting and help to achieve better robustness.
- All the experiments are performed for eps=8/255 which is close to the eps for which even standard FGSM leads to high robustness (e.g., see Fig. 1 in Andriushchenko & Flammarion (2020)). Thus, it would be highly beneficial to show that the proposed approach works equally well compared to the competing approaches for a higher eps, e.g. eps=16/255 on CIFAR-10.
- The proposed procedure of backward smoothing already appears in the literature in [Diversity can be Transferred: Output Diversification for White- and Black-box Attacks](https://arxiv.org/abs/2003.06878) where they propose a very similar algorithm but with the goal of obtaining a better adversarial attack (and thus they use multiple steps of PGD to solve it instead of just one). Although this paper is not yet published (accepted at NeurIPS’20), I think it would be important to at least acknowledge this connection.
- I am still quite surprised that for the inner maximization problem, taking two steps of PGD wrt *different* losses leads to higher robustness compared to taking two steps of PGD wrt the same original loss function. In order to rule out potentially suboptimal choice of the step size for *Fast TRADES (2-step)*, I would be interested in seeing results of a grid search over it, i.e. similarly to Table 7 but for *Fast TRADES (2-step)*.


**Minor suggestions:**
- Fig. 2: the best step size is at the boundary of the grid {10/255, 8/255, 7/255}, thus it would make sense to extend the grid with even smaller values of the step size to make sure that Fast TRADES is not reported with suboptimal hyperparameters.
- I'm also a bit surprised that *"Backward smoothing ... even outperforms the state-of-the-art robust training methods"* since the SOTA robust training methods include much more steps of PGD and thus they solve better the inner maximization problem. Why is their performance worse? Can it be, for example, because their hyperparameters were not sufficiently tuned (unlike for proposed method)?
- Table 5: the listed "AT" baseline is quite weak (44.04%) from Madry et al. (2018). It would be better to use the results / models from Rice et al. (2020) which show results of AT+early stopping comparable to that of TRADES.
- Table 5: it would be useful to add *Fast TRADES (2-step)* to the table as this is the most interesting baseline.


**Score:**
My current score is 4/10 but I would be willing to raise it if my first two concerns (mentioned in **Cons**) are resolved.

------

**Update after the public discussion:**

Thanks a lot to the authors for clarifying many details and providing additional experimental data. In the updated version of the paper, the authors improve the results of the baseline "Fast TRADES (2-step)" and add additionally a stronger baseline of "Fast AT (2-step)" (except on CIFAR-10 with eps=16/255 where it's missing). However, at least for eps=8/255 on CIFAR-10, CIFAR-100, Tiny ImageNet, the authors show consistent improvements over the baselines with comparable computational cost ("Fast TRADES (2-step)" and "Fast AT (2-step)"). This is an interesting result, although it's not clear to me whether it's specific to eps=8/255 or it would generalize also to higher epsilons such as eps=12/255 or eps=16/255.

On the other hand, I still think that the current motivation of the method is very incomplete and it is still unclear why the proposed method should work in the first place. Perhaps, it's a good idea to further develop the additional experiments about the curvature of the loss surface at different points in the input space.

Then I think there is additional work to be done in terms of understanding what the proposed method actually does (even if we don't take into account how it was motivated -- via randomized smoothing or not). In particular, it's still completely unclear to me why 2 steps of PGD with respect to the original KL divergence (i.e. Fast TRADES (2-step)) works worse than first 1 step with respect to one KL-divergence and then 1 step with respect to another KL-divergence (i.e. Backward Smoothing). This seems quite ad-hoc and requires further explanations, in my opinion. Moreover, I find it also quite puzzling that Backward Smoothing even outperforms 10-step TRADES / AT as shown in Tables 3 and 4 -- not sure about a justification behind this.

Taking into account all of this, I update my score from 4/10 to 5/10. I think the paper can still be improved in various ways: both in terms of the motivation and experimental results.

---

> ### Author Response · Authors · 2020-11-17
> **Authors’ Response to Reviewer 2**
>
> Thank you for your insightful comments. Below, we provide detailed responses to your questions.
>
> Q1: The explanations from Section “Why random initialization helps?” are not convincing ...This is my main concern... achieve better robustness.
>
> A1: First, please refer to our response to all reviewers regarding the relationship with the pointed paper. Second, using a step size of 2*eps is simply too large. To the best of our knowledge, there does not exist a single-step adversarial training algorithm that works with step size 2*eps. Nevertheless, we can show that we indeed take the step size into account and the comparison with Fast AT on various step sizes shows the advantage of our proposed method.
>
> | Step Size | Fast AT | Backward Smoothing |
> |-----------|---------|--------------------|
> | 6/255     | 41.47   | 52.38              |
> | 7/255     | 43.00   | 52.40              |
> | 8/255     | 44.46   | 52.50              |
> | 9/255     | 45.39   | 52.16              |
> | 10/255    | 46.30   | 52.04              |
> | 11/255    | 39.66   | 42.45              |
> | 12/255    | 38.52   | 41.12              |
>
> Q2: All the experiments are performed for eps=8/255... compared to the competing approaches for a higher eps, e.g. eps=16/255 on CIFAR-10.
>
> A2: Thank you for your suggestion, we have added experiments with eps=16/255 for CIFAR-10 dataset on the ResNet18 model.
>
> | Method               | Nat (%) | Rob (%) | Time (min) |
> |----------------------|---------|---------|------------|
> | AT                   | 62.76   | 32.03   | 425        |
> | Fast AT              | 53.72   | 20.12   | 89         |
> | TRADES               | 62.09   | 28.63   | 470        |
> | Fast TRADES          | 56.55   | 17.47   | 137        |
> | Fast TRADES (2-step) | 57.22   | 19.28   | 167        |
> | Backward Smoothing   | 63.47   | 25.04   | 164        |
>
> We can see that Backward Smoothing still achieves significant improvements over other single-step adversarial training algorithms. This verifies the effectiveness of the proposed method.
>
> Q3: The proposed procedure of backward smoothing... I think it would be important to at least acknowledge this connection.
>
> A3: Thank you for pointing it out. We will acknowledge the connection and discuss the paper in the revision.
>
> Q4: For the inner maximization problem, taking two steps of PGD ... results of a grid search over it, i.e. similarly to Table 7 but for Fast TRADES (2-step).
>
> A4: Thank you for your suggestion, we have added grid search results for Fast TRADES (2-step).
>
> | Step Size | Fast TRADES |
> |-----------|-------------|
> | 4/255     | 48.54       |
> | 5/255     | 48.78       |
> | 6/255     | 48.33       |
> | 7/255     | 48.08       |
> | 8/255     | 48.01       |
> | 9/255     | 47.38       |
> | 10/255    | 45.87       |
>
> We can observe that after careful tuning, we manage to improve the robustness of Fast TRADES (2-step) by 0.3%. We will update the result in the revision.
>
> Q5: Fig. 2: the best step size is at the boundary of the grid {10/255, 8/255, 7/255}, ... Fast TRADES is not reported with suboptimal hyperparameters.
>
> A5: Thank you for your suggestion, we also added extra tuning results for Fast TRADES.
>
> | Step Size | Fast TRADES |
> |-----------|-------------|
> | 5/255     | 44.01       |
> | 6/255     | 45.91       |
> | 7/255     | 45.21       |
> | 8/255     | 43.36       |
> | 10/255    | 38.83       |
>
> We observe that a step size of 6/255 achieves slightly better robustness (0.7%) for Fast TRADES. We will update the result in the revision.
>
> Q6: I'm also a bit surprised that ...Can it be, for example, because their hyperparameters were not sufficiently tuned (unlike for proposed method)?
>
> A6: First, we did not claim that backward smoothing constantly beats the state-of-the-art methods, we only achieve slightly better results for one or two settings. And we indeed tuned the hyperparameters for all methods. Second, the success of Fast Adversarial Training has already suggested that for adversarial training, stronger attacks for the inner problem are not required as mentioned in their original paper.
>
> Q7: Table 5: the listed "AT" baseline is quite weak (44.04%) ... show results of AT+early stopping comparable to that of TRADES.
> Table 5: it would be useful to add Fast TRADES (2-step) to the table as this is the most interesting baseline.
>
> A7: Thank you for your suggestion, we have retrained an AT model with early stopping and also tested Fast TRADES (2-step) for Table 5.
>
> | Method               | AutoAttack |
> |----------------------|------------|
> | AT                   | 44.04      |
> | AT (early-stop)      | 49.10      |
> | Fast AT              | 43.21      |
> | TRADES               | 53.08      |
> | Fast TRADES (2-step) | 48.21      |
> | Backward Smoothing   | 51.13      |

---

> > ### Comment · AnonReviewer2 · 2020-11-24
> > **Still not convinced about the role of randomness and the overall motivation of the method**
> >
> > Thanks a lot for the additional experiments.
> >
> > *"Second, using a step size of 2eps is simply too large. To the best of our knowledge, there does not exist a single-step adversarial training algorithm that works with step size 2eps."*
> >
> > I'm not sure what *"simply too large"* means. There exist single-step methods that work with step size 2 eps: e.g. see GradAlign+FGSM or CURE+FGSM reported in Andriushchenko & Flammarion (2020) that work fine with a full step size of 2eps (i.e. when the adversarial perturbations are in $\\{-\varepsilon, \varepsilon\\}^d$ if we ignore the box constraints of images $\\{0, 1\\}^d$) even for eps=16/255.
> >
> > So my conclusion is that randomness is still not relevant to the success of fast adversarial training (or, at least, there is no evidence to claim that it is).
> >
> > -----
> >
> > *"we have added experiments with eps=16/255 for CIFAR-10 dataset on the ResNet18 model."* and *"we did not claim that backward smoothing constantly beats the state-of-the-art methods, we only achieve slightly better results for one or two settings"*
> >
> > Andriushchenko & Flammarion (2020) report ~28% adversarial accuracy for adversarial training with PGD-2 on ResNet-18 for eps=16/255 (when early stopping is used, see Fig. 14 therein). One of my major concerns is that this approach may be just on par with 2-step PGD adversarial training when evaluated carefully and thus providing no extra benefits. I would be quite hesitant to recommend acceptance of a considerably more involved procedure if there exists a simpler and more established alternative of simple adversarial training with fewer steps.
> >
> > -----
> >
> > *"Second, the success of Fast Adversarial Training has already suggested that for adversarial training, **stronger attacks for the inner problem are not required** as mentioned in their original paper."*
> >
> > I'm not convinced about this claim. The gap between the robustness of multi-step PGD training and one- / two-step PGD training is growing over the size of the $\ell_\infty$ epsilon. In fact, the difference is noticeable even for $\varepsilon=8/255$ on CIFAR-10 as Wong et al. (2020) report, although they comment on this differently.
> >
> > -----
> >
> > Regarding the new experiment with the maximal eigenvalue of the Hessian: I think it's indeed an interesting experiment that shows that the curvature at a backward smoothed point is somehow smaller compared to the curvature at a random point or at the original input. At the same time, this experiment seems to invalidate the connection between the proposed backward smoothing and randomized smoothing as according to the new Figure 3, from the Hessian point of view, there is no significant difference on whether to train on original or randomly perturbed points.
> >
> > -----
> >
> > To conclude, I would stay with my original opinion. I am still (1) not convinced that randomness has any important role and (2) the connection between randomized smoothing and backward smoothing is questionable. There is some empirical success of the proposed scheme, but in my opinion it's not sufficient. So I think the paper requires a revision to better motivate the method (particularly, the connection to randomized smoothing or the new perspective on the curvature) and more clearly show that it outperforms 2-step PGD adversarial training (not just 2-step TRADES) over multiple Linf epsilons.

---

> > > ### Author Response · Authors · 2020-11-25
> > > **Thank you for your reply**
> > >
> > > 1. “There exist single-step methods that work with step size 2 eps... that work fine with a full step size of 2eps… ”
> > >
> > > - This is a huge misunderstanding. We have carefully checked the hyperparameter configuration of GradAlign in (Andriushchenko & Flammarion, 2020), the authors only claimed that GradAlign works with *perturbation constraint* epsilon=8/255 as well as 16/255, rather than using a *step size* equal to 2eps. Their public codebase also suggests that they use a step size of 1.25eps, which is the same as Fast AT.
> > >
> > > 2. “randomness is still not relevant to the success of fast adversarial training. ”
> > > - We respectfully disagree. The effect of randomness in Fast Adversarial Training is evident since without randomness, FGSM training with a step size of eps will fail as demonstrated in the Fast AT paper.
> > >
> > > 3. “One of my major concerns is that this approach may be just on par with 2-step PGD adversarial training when evaluated carefully and thus providing no extra benefits”
> > > - The ~28% robust accuracy reported in (Andriushchenko & Flammarion, 2020) is based on PreActResnet-18, while our additional experiments are based on the Resnet-18. Therefore, our results and theirs are not directly comparable.  We will carry out an additional experiment of your suggested baselines for eps=16/255 setting on the Resnet-18, and add it to the final version. Due to the short notice, we are afraid that we are not able to finish these experiments by the author response deadline.
> > >
> > >
> > > 4. “I'm not convinced about this claim.”
> > > - It is actually *not* our claim but Fast AT paper’s claim (section 5.5 point 2 said, “Defenders don’t need strong adversaries during training”). Also we did not claim that our method could achieve the same robustness as multi-step robust training, we are claiming that it could achieve similar performances, at least for our proposed method.
> > >
> > > 5. “from the Hessian point of view, there is no significant difference on whether to train on original or randomly perturbed points”
> > > - As can be seen from Figure 3, the maximum Hessian eigenvalue of random perturbed points (orange curve) is significantly lower than that of the original points (blue curve). This suggests that randomized smoothing indeed makes the loss function smoother than the original loss. In addition, the maximum Hessian eigenvalue of perturbed points by Backward smoothing (green curve) is drastically lower than that of random perturbed points (orange curve), which strongly supports the improvement of Backward Smoothing over randomized smoothing.

---

### Official Review · AnonReviewer4 · 2020-10-29
**Insufficient motivations and the improvements seem to be a trade-off.**

**Rating:** 5
**Confidence:** 4

**Review:**

This work proposes backward smoothing as an advanced random initialization to improve a model's adversarial robustness. The paper is well-written and easy to follow. However, I have the following concerns:
1) The paper argues that random initialization can help fast adversarial training because it helps improve the smoothness of the loss function. However, a recent work "Understanding and Improving Fast Adversarial Training" has a very nice explanation and solution on this problem, which is much more convincing to me.
2) Even if I accept the condition that random initialization helps improve the smoothness of the loss function, the motivation showing in Figure 3 that the current smoothing effect by using random initialization is not sufficient can not convince me. It is definitely in the expectation that KL divergence between clean and adversarial examples would be much greater than random initialization. And I do not think that this can shed any light on the insufficient smoothness by the current random initialization.
3) At the end of section 4, the authors mention that "Note that the proposed Backward Smoothing seems also compatible with Adversarial Training. However, Adversarial Training does not contain terms using KL divergence loss, which may hinder its performance. We will show this empirically in Section 5." I tried to find experiments with backward smoothing applied to standard adversarial training but did not find these results. It would be good if the authors can point it to me if I missed them. Conditioned on  that we observe backward smoothing does not greatly help standard adversarial training, this failure case raises a great concern to me. Compared to random initialization, backward smoothing provides a better random initialization which can provide better smoothness. My question is: why a better random initialization can not generalize to standard adversarial training?
4) For Table 5, it is necessary to include FAST TRADES as backward smoothing is mainly applied to TRADES.
5) Does backward smoothing really help adversarial robustness or it just plays the tradeoff game between clean accuracy and adversarial robustness? For example in Table 8, comparing FAST Trades and backward smoothing, backward smoothing provides better adversarial robustness but sacrifices a big clean accuracy.

In all, I vote for a reject for the current version of this work.

********After Discussion*************
I thank the authors for answering my questions and performing additional experiments. Some of my concerned are addressed during the discussion stage. Therefore, I raise my score from 4 to 5. However,  I still hold my opinions that this work does not have a strong motivation, does not help too much for standard adversarial training and has a potential trade-off problem.

---

> ### Author Response · Authors · 2020-11-17
> **Authors’ Response to Reviewer 4**
>
> Thank you for your insightful comments. Below, we provide detailed responses to your questions.
>
> Q1: (Andriushchenko & Flammarion, 2020) has a nice explanation and solution, which is more convincing to me.
>
> A1: See our comments to all reviewers.
>
> Q2: Even if I accept the condition..., the motivation showing in Figure 3 ... is not sufficient and can not convince me... current random initialization.
>
> A2: We have also added a new experiment in Figure 3 in the revision to better show that smoothing indeed plays an important role in training robust models and the current smoothing effect is not enough. Specifically, we show the maximum eigenvalue of Hessian of the loss function at the original examples, randomly perturbed examples, and backward smoothed examples along the training trajectory in Figure 3. The result suggests that randomized smoothing indeed helps make the loss function smoother yet it is not as effective as our proposed backward smoothing technique. We hope this new experiment can address your concern.
>
> Q3: At the end of section 4, the authors mention …but did not find these results… why a better random initialization can not generalize to standard adversarial training?
>
> A3: In Section 5.4, we analyzed the compatibility of our algorithm with Adversarial Training. And as we explained in Section 5.4 (starting from line 6), we think the main reason for the deteriorated performance is the different choices of inner maximization loss for Adversarial Training and TRADES. Considering the random perturbation generated on the output space, the Cross-Entropy loss mainly focuses on the $y$-logit (limited smoothing effect) while KL divergence is closely related to all logits.
>
> Q4: For Table 5, it is necessary to include FAST TRADES as backward smoothing is mainly applied to TRADES.
>
> A4: Thank you for your suggestion. We have added the FAST TRADES result in Table 5.
>
> | Method             | AutoAttack |
> |--------------------|------------|
> | AT                 | 44.04      |
> | Fast AT            | 43.21      |
> | TRADES             | 53.08      |
> | Fast TRADES        | 43.85      |
> | Backward Smoothing | 51.13      |
>
> Q5: Does backward smoothing really help adversarial robustness or it just plays the tradeoff game ...but sacrifices a big clean accuracy.
>
> A5: First, in all experiments and for all methods, we have tuned the models for their best robustness performances. Second, notice that our method is based on TRADES, and compared with TRADES, backward smoothing actually achieves very similar clean accuracy as well as robust accuracy, suggesting that we are not playing the tradeoff game. On the other hand, baselines such as Fast TRADES and Fast TRADES (2-step) are actually sacrificing robustness for better natural accuracy (compared to TRADES). This is why we need backward smoothing to fix this.

---

> > ### Comment · AnonReviewer4 · 2020-11-24
> > **Still not convinced by the motivation and trade-off performance**
> >
> > Thanks very much for the additional experiments and explanations.
> >
> > First, I still think Andriushchenko & Flammarion, 2020 gives a better and more fundamental explanation on how to address fast adversarial training after I looked through the authors' response.
> >
> > Second, backward smoothing does not generalize to standard adversarial training. In other words, it only works for KL divergence loss rather than cross-entropy loss, this is still a big concern to me and it seems that backward smoothing does not address the fundamental problem of fast adversarial training.
> >
> > Third, I do not think the authors well address my concern about the trade-off problem. If we simply compare the performance of Fast TRADES and Backward Smoothing since Backward Smoothing intends to improve Fast TRADES, there is always a tradeoff between clean accuracy and adversarial robustness, e.g., Table 4 and Table 8.
> >
> > In all, I would keep my rating unchanged.

---

> > > ### Author Response · Authors · 2020-11-25
> > > **Thank you for your reply**
> > >
> > > 1. “First, I still think Andriushchenko & Flammarion, 2020 gives a better and more fundamental explanation…”
> > >
> > > - We respectfully disagree with you. As we explained before, our explanation based on smoothness is more general and can cover theirs. Can you explain why theirs is more fundamental?
> > >
> > > 2. “it only works for KL divergence loss rather than cross-entropy loss… it seems that backward smoothing does not address the fundamental problem of fast adversarial training”
> > >
> > > - Our method works for both, and it works better for KL divergence than for cross-entropy loss. Also, when it is applied to KL divergence, it gives the best-known performance and beats all previous methods. Could you clarify what is the “fundamental problem of fast adversarial training” you are referring to?
> > >
> > > 3. “I do not think the authors well address my concern about the trade-off problem”
> > >
> > > - Both Fast TRADES and Backward Smoothing are proposed in this paper. While Fast TRADES achieves higher natural accuracy, Backward Smoothing achieves higher robust accuracy. The tradeoff you are concerned with is between two of our proposed methods. But given that one of these two methods, Backward Smoothing, achieves comparable performance as TRADES in both natural accuracy and robust accuracy, while being much faster than TRADES, we don’t think the tradeoff you’re talking about will downgrade our contribution.

---

### Author Response · Authors · 2020-11-17
**General Response to All Reviewers**

Reviewers 1,2,4 all mentioned an alternative explanation is presented in (Andriushchenko & Flammarion, 2020), we also read and cited this work in our submission. They provided an explanation of why Fast Adversarial Training works: random initialization reduces the magnitude of the perturbation and thus the network becomes more linear and fits better toward FGSM attack. While we argue that the random initialization works as randomized smoothing for smoothing the inner maximization problem (making it easier to solve). In fact, our argument is more general and can cover theirs, because if the loss function is approximately linear, then it will be very smooth (i.e., the second-order term in the Taylor expansion is very small). Also note that our method indeed achieves very close performance with current-best methods on benchmark datasets and leaderboards. We will discuss the relationship of the two works in detail in our revision.

---

### Author Response · Authors · 2020-11-23
**New Experiments (Figure 3) Added**

Reviewers 3 and 4 raised concerns about the role of smoothing in robust training, e.g., Figure 3 in the original submission is not informative. We have now added a new experiment replacing old Figure 3 in the revision to better show that smoothing indeed plays an important role in training robust models and the current smoothing effect is not sufficient. Specifically, we show the maximum eigenvalue of Hessian of the loss function at the original examples, randomly perturbed examples, and backward smoothed examples along the training trajectory in new Figure 3. The result suggests that randomized smoothing indeed helps make the loss function smoother yet it is not as effective as our proposed backward smoothing technique. Please refer to the updated pdf file for more details.

---

### Decision · Program_Chairs · 2021-01-07
**Final Decision**

**Decision:**

Reject

**Comment:**

This paper studies efficient robust training. The key idea is to use backward smoothing as an advanced random initialization to improve a model's adversarial robustness. The approach is sound, well grounded, and quite logical. Results demonstrate the effectiveness.

However, there exists some limitations:

1) Andriushchenko & Flammarion, 2020 gives a better and more fundamental explanation on how to address fast adversarial training.

2) Backward smoothing does not generalize to standard adversarial training. In other words, it only works for KL divergence loss rather than cross-entropy loss, and it seems that backward smoothing does not address the fundamental problem of fast adversarial training.

3) If we compare the performance of Fast TRADES and Backward Smoothing since Backward Smoothing intends to improve Fast TRADES, there is always a tradeoff between clean accuracy and adversarial robustness, e.g., Table 4 and Table 8.

4) Randomized smoothing is helpful for one-step adversarial training and randomized smoothing in general seems to be orthogonal to the proposed method. Moreover, 2-step PGD training can perform similarly well to backward smoothing while being much simpler conceptually.

In the end, I think that this paper may not be ready for publication at ICLR, but the next version must be a strong paper if above limitations can be well addressed.